# Parent’s Hesitation with COVID-19 Vaccinations in Infants and Children Aged 6 Months to 5 Years

**DOI:** 10.3390/vaccines10111828

**Published:** 2022-10-29

**Authors:** Austin M. Temple, Evelyn Schendler, John Harrington

**Affiliations:** 1Eastern Virginia Medical School (EVMS), 825 Fairfax Ave, Norfolk, VA 23507, USA; 2Children’s Hospital of The King’s Daughters (CHKD), 601 Children’s Lane, Norfolk, VA 23507, USA

**Keywords:** immunization, FDA, EUA

## Abstract

We implemented an in-person survey of parents/guardians concerning COVID-19 vaccine for a predominantly African-American Medicaid pediatric patient population between the ages of 6–59 months at a Children’s Hospital General Pediatric Clinic in Norfolk, VA. Vaccine hesitancy was predominantly based on concerns surrounding safety and overall need for the vaccine.

## 1. Introduction

The age-based longitudinal implementation of COVID-19 immunizations has seen a negative correlation of vaccination rates among the sequentially younger age groups. Over 80% of individuals over age 50 have completed a COVID-19 immunization series compared to only 30% of individuals aged 5 to 11 years [1]. With FDA Emergency Use Approval of COVID-19 vaccinations for children aged 6 months to 5 years, it is likely we will see the same downward trend in vaccination rates among this youngest approved age group. In fact, a recent internet survey by Scherer et al. found that only 20% of parents intend on getting their child vaccinated within the next three months [2].

Generally, children 6 to 59 months experience less side effects and symptoms during and after COVID-19 infection relative to older age groups [3]. However, systematic reviews of COVID-19 for children under five years of age have also shown mild to moderate symptoms for those infected, with a significant proportion of those individuals being infants [4]. The assumed lower rates of severity in younger age groups seems to have contributed to the belief that vaccination is unnecessary for these age groups or that vaccine may be unnecessary after natural COVID-19 infection.

New variants have also posed increased risks to children in these age groups and have led to higher rates of emergency room visits and hospitalizations [5]. For children hospitalized with an omicron strain of COVID-19, an increase in the number moderate cases was observed relative to the number of moderate cases with pre-Omicron infections, but an overall lack of severity of cases was noted. Incidence of COVID-19 infections has continued to increase in the 6–59-month population due to lack of eligibility for vaccines and greater infectivity among newer variants [5]. Children aged 0–24 months demonstrated the highest rates of Omicron infection versus 25–48 month age groups in recent studies [6].

Even with lower rates of severe illness in children, they are still suffering from long-term complications. Children recently hospitalized with the Omicron strain of SARS-CoV-2 were more likely to suffer from an acute upper airway infection (UAI) than those hospitalized with a pre-omicron strain (4.1% versus 1.5%). Those who suffered from COVID-19 induced UAI had a higher chance of developing severe symptoms relative to those who did not develop an UAI. The presence of UAI is linked to increased, but still rare rates of cardiac arrest and other long-term health problems [7]. Multisystem Inflammatory Syndrome in Children (MIS-C) is another rare but serious systemic manifestation of COVID-19 infection with an incidence rate of 2 per 100,000 cases. Cardiovascular impairment, respiratory, GI and neurocognitive symptoms may all be present in previously healthy children who present with COVID-19 infection [8].

Within many families, young children are most likely, among all age groups, to have closest and prolonged types of contact with respiratory secretions of other kids. For many, school is in person and has variable masking in place. A lack of immunization and increase in social activity can lead to higher rates of childhood infection with COVID-19 along with increased transmission to family members who may be unable to receive vaccination themselves. Finally, vaccination of this youngest age will decrease COVID transmission rates [9].

With these concerns, we sought to identify specific reasons for COVID vaccine hesitancy and hope to create a targeted plan to improve vaccination rates in this vulnerable age group.

## 2. Materials and Methods

From May 31st through July 5th, 2022, individuals presenting to the Children’s Hospital of The King’s Daughters General Academic Pediatrics (CHKD GAP) clinic were approached by a research assistant upon entrance into the clinic. The CHKD GAP clinic is primarily a safety net pediatric practice with a large majority of patients on Medicaid or Medicaid managed care in Norfolk, VA. The research assistant assessed the individual’s eligibility to participate in the survey, informed the individual of the purpose of the study, and obtained consent. Inclusion criteria included individuals with non-COVID-19 vaccinated children aged 6–59 months. Individuals who agreed to participate were given a 23-question multiple choice survey that assessed their willingness to vaccinate their children based on demographic information, existence of pre-existing conditions, personal beliefs, and possible motives, and where they receive their information concerning vaccines. The survey provided 18 pre-selected answer choices and participants were instructed to chose their top 3 sources for information about COVID-19 vaccines. A copy of the survey is available in the Appendix A. Individuals with more than one child within the 6–59-month age group who were not COVID-19 vaccinated were instructed to only complete one survey and use the information for the youngest qualifying child. In the case of a qualifying family returning to the clinic on another date, they were not asked to participate again and their future attendance was not counted. In cases where eligibility status could not be attained or translation services were unable to be provided, individuals were marked as ineligible and not counted towards the total. The distributed survey was available in English without translations offered. If the individual was not proficient in English and did not feel comfortable completing the survey, it was not distributed. 40 surveys were not distributed due to the participant not confident in their ability to answer the English-language survey. Surveys were collected by a member of the CHKD GAP clinic staff or a research assistant directly after the visit. Surveys completed with a signed consent were entered into the database repository called REDCAP for data analysis. Surveys that were not completed or had no signed consent were not entered into REDCAP, but were counted to determine the total number of surveys distributed. Individuals who were assessed to be eligible but denied participation before a survey could be distributed were also counted to determine the eligible population of the CHKD GAP clinic. Chi-Square Goodness of Fit tests and Fisher’s Exact tests were utilized to analyze associations between variables. All statistical tests were performed using SPSS.26 (Chicago, IL) and were two-sided with *p* < 0.05 being considered statistically significant. Logistic regression model was also used to calculate Odds Ratio and 95% Confidence Interval.

## 3. Results

204 surveys out of 336 distributed surveys to eligible participants were consented and completed. In total 375 individuals were assessed to be eligible but 39 individuals denied participation before a survey could be distributed. Responses showed equal representation of age groups within the 6 months to 4-year range with a racial mix of 65% African American, 20% Caucasian, 5% Hispanic and 5% multi-racial and Asian and most with a yearly income under $50,000. 53% of caretakers of qualifying children assessed where between the age of 25 and 34, with 96% being between the age of 18 and 44. 42% of individuals attested to completion of high school or General Educational Development test (GED), 22 % completed some college or trade school, while 33% had completed either a 2 or 4 year college degree plan. A statistically significant association between age groups and intent to vaccinate was not found (*p* = 0.58) (Figure 1).

Individuals were divided into two groups ‘unlikely to vaccinate’ (N = 131) and ‘likely to vaccinate’ (*n* = 60) based on their responses within the survey. Individuals who marked that they were ‘unlikely to ever vaccinate’ or ’unlikely but may consider in the future when [they] know more’ were grouped into the ‘unlikely to vaccinate’ category. Individuals who marked ‘unlikely today but will do it in the future from what I know now’, ‘likely would do it today if I had time so will schedule later’, ‘very likely to do it today if available’ and ‘my child has received the vaccine’ were grouped into the likely to vaccinate’ cohort. Sources of hesitancy were assessed and compared between the two groups. A majority (78/131) of individuals who were ‘unlikely to vaccinate’ responded that they needed more information to make an informed choice. 27% (36/131) of these same individuals believed that the vaccine was more likely to harm their child than help them and 29% (38/131) wanted to see more children immunized first before immunizing their own. There was a lack of trust in the establishment of government/healthcare with increased responsiveness by the ‘unlikely to vaccinate’ group (28/131). When grouped by ethnicity those identifying as Caucasian were significantly more likely to vaccinate (51% or 19/37) compared to African-Americans (26% or 32/109, *p* = 0.007). Due to limitations in sample size, comparisons could not be made for other groups (Figure 2).

Parental concern for their child catching COVID-19 was assessed for the ‘unlikely to vaccinate’ and ‘likely to vaccinate’ groups (Figure 3). There was a statistically significant difference between the two groups, with the ‘unlikely to vaccinate’ group displaying a bimodal concern for their child’s COVID-19 susceptibility (*p* = 0.012). Individuals who were more concerned about their child catching COVID-19 were more likely planning to vaccinate their children in the future (*p* = 0.012). There was no statistical difference for the concern of one’s child catching COVID-19 between African-American and Caucasian groups (*p* = 0.41).

Possible future motivating reasons (i.e., would allow vaccination to protect the health of my child/friends/family/community, to get back to travel/school/work) were evaluated to see what could motivate a hesitant parent to agree with vaccination. The presence of a positive motivational reason in a parent/guardian was significantly higher when they were ‘likely to vaccinate’ relative to those who were unlikely. 50% (66/131) of ‘unlikely to vaccinate’ individuals would allow vaccination if it meant protection of their child’s health. Individuals who were ‘likely to vaccinate’ were more likely to allow vaccinations in cases where it allowed for resumed travel (30% ‘likely to vaccinate’ vs. 7.6% ‘unlikely to vaccinate’), resume social activities (29% vs. 11.5%), protect the health of friend and family (61.3% vs. 22.9%), protect the health of my child’s friend’s and peers (51% vs. 18%) and protect the health of the community (50% vs. 13%). When prompted on would a discussion with a physician or family member help to influence their vaccination decision 17% (23/131) of ‘unlikely to vaccinate’ individuals would like more time to discuss with their physician and 13% (17/131) of the same group would like more time to discuss with their family. Individuals who were more likely to vaccinate their children were more likely to believe that the vaccine was safe and effective (*p* < 0.001). ‘Likely to vaccinate’ individuals were also more likely to believe that they did not need more information to make a informed choice for vaccination (*p* < 0.001). Parents who were ‘unlikely to vaccinate’ were less likely to have trust in the establishment of government/healthcare (*p* = 0.003).

Parental belief in vaccine safety is described in Figure 4. Individuals who were ‘unlikely to vaccinate’ had greater levels of belief that the vaccine was not safe for their child (*p* < 0.001). 50% of ‘unlikely to vaccinate’ individuals believed that the vaccine was ‘not safe at all’ with 35% believing that it was ‘a little safe’ or ‘moderately safe’. This was compared to 86% of ‘likely to vaccinate’ individuals believing that the vaccine was ‘moderately safe’ or ‘very safe’. African-Americans were more likely to perceive the vaccine as less safe (64% or 76/118) compared to Caucasian guardians (40.5% or 15/37, *p* = 0.011). Sources of information were analyzed to determine if parent’s felt as if they were receiving too much or not enough information in regards to COVID-19 vaccination. Overall 65% of parents listed the Centers for Disease Control and Prevention (CDC) as one of their top 3 most trusted news sources. Hospital system websites followed second with 52% of parents endorsing trust and local health officials were 3rd most listed with 27%. These sources were consistent amongst the ’too much information’ and ’not enough/just enough information’ groups.

Other factors analyzed to determine correlations were calculated. There was no statistical significance between parental concern of vaccine safety and their child catching COVID-19 (*p* = 0.14). There was no statistical significance between the likelihood of planning to vaccinate and prior COVID-19 exposure (*p* = 0.88). Nor was there a statistical significance between knowledge of a closely related friend or relative who was seriously ill or had died due to COVID-19 (*p* = 0.56). In addition, there was no statistical significance between the likelihood of receiving a COVID-19 vaccine and having individuals with pre-existing conditions within the household (*p* > 0.05).

## 4. Discussion

The results of this study provide key insights for healthcare providers to increase vaccination rates for both COVID-19 and future variant vaccines. Our results mirror recently published articles describing high rates of parental hesitancy when questioned on future vaccination plans [2]. Vaccine hesitancy was based primarily on a parent’s belief that the vaccine is unsafe and that their child has little risk of getting sick from a natural COVID-19 infection.

An important trend was that parents who were ‘unlikely to vaccinate’ still had high levels of concern over their child catching COVID-19. This could demonstrate levels of parental belief that vaccination has greater health detriments than COVID-19 infection itself. The risk of developing myocarditis in pediatric populations after COVID-19 immunization is higher than the development of myocarditis in the non-immunized population [10]. A study by Oster et al. suggests that the rate of developing myocarditis is less than 0.0001% while hospitalization rates of myocarditis for those under the age of 30 with detailed clinical information available was close to 96%. This is in comparison to CDC data estimating over 9000 cases of MIS-C induced by COVID-19 infection requiring hospitalization in children in the United States [11]. From December 2021 through February 2022, there were 397 children hospitalized due to COVID-19 within the United States. Of that number 87% were unvaccinated and 19% were eventually admitted to the ICU. Unvaccinated children were 2.1 times more likely to be hospitalized relative to vaccinated children [12]. Protection and safety for their child was a large factor in vaccine compliance. With both likely and unlikely to vaccinate groups displaying similar numbers of individuals who would allow vaccination if they believed it would protect their child. Parents who are ’unlikely to vaccinate’ may not be susceptible to the same type of information that pro-vaccination group are. This lack in effective messaging may be one of the hurdles that has yet to be addressed in this population. Once parents believed that vaccines were safe and effective, their motivation to vaccinate increased. Parents who demonstrated agreement with motivational reasons were more likely to belong to the ‘likely to vaccinate’ cohort. By appealing to these motivational reasons, discussing the safety and efficacy of vaccination, and taking the time to discuss concerns with parents, healthcare professionals should be able to increase vaccination rates among the hesitant populations. Generally, parents who were ’unlikely to vaccinate’ were less likely to vaccinate their child if it meant doing so to increase community protection or protect the health of friends and family. This could be attributed to a lack of belief that vaccination of children would lead to an increase community protection/public health or that vaccination was inherently more harmful to a child relative to the community benefits it may possess.

Importantly, hesitant parents showed willingness to talk with their physicians about vaccinations and were open to discussing the benefit of vaccinations on their child’s well being. The ability for the physician to sit down with the individual and answer questions may provide the persuasive opportunity to convince uncertain parents about vaccinations. A large proportion of individuals still saw a physician as an important figure within their medical decision making process and would welcome the opportunity to openly discuss concerns in a non-judgemental setting. With increased sources of information and an increased prevalence of diffracting opinions within those sources, parents look towards physicians to provide impartial and evidence-based information.

A difference in the perceptions of vaccination by African-American individuals and Caucasian individuals was observed. African-American individuals had lower rates of confidence in the COVID-19 vaccine and demonstrated a lower intent to vaccinate compared to Caucasian individuals. Yet, both groups had similar levels of concern for their child’s susceptibility to infection. Determining the root of this contrast is important to increasing vaccination rates in more vulnerable populations.

Future studies should account for the relatively small number population we were sampling from within the CHKD GAP clinic. A 60% response rate in distributed surveys should not be discounted, but an increase in population would only strengthen the conclusions made. A larger sample pool, or data sourced from more locations will help to increase the statistical significance of what was collected. A follow-up study assessing these same parameters would be beneficial, now that a vaccine for the 6–59 month age group is approved and in circulation. Would the increase in children being vaccinated lead to motivations for parent’s who were previously undecided? Would we be seeing the same gradual plateau in vaccination rates that have been present with all of the other approved age groups. Unfortunately, we also did not ask specific questions related to the current vaccine not containing the predominant variants circulating and the need to give 2–3 doses to achieve partial transient immunity, which from the provider’s view may be awkward to explain since the likelihood of future bivalent or antigen specific vaccines may be required without full vaccine trials.

The message we do hope to convey by this study is to show that when parents believe that the vaccine is safe and effective, they will most likely proceed with vaccination. The goal is to not frighten parents, but instead appeal to their personal motivations so that they will want to make the choice based on the best evidence available and provided by a trusted source in a non-threatening way.

## Figures and Tables

**Figure 1 vaccines-10-01828-f001:**
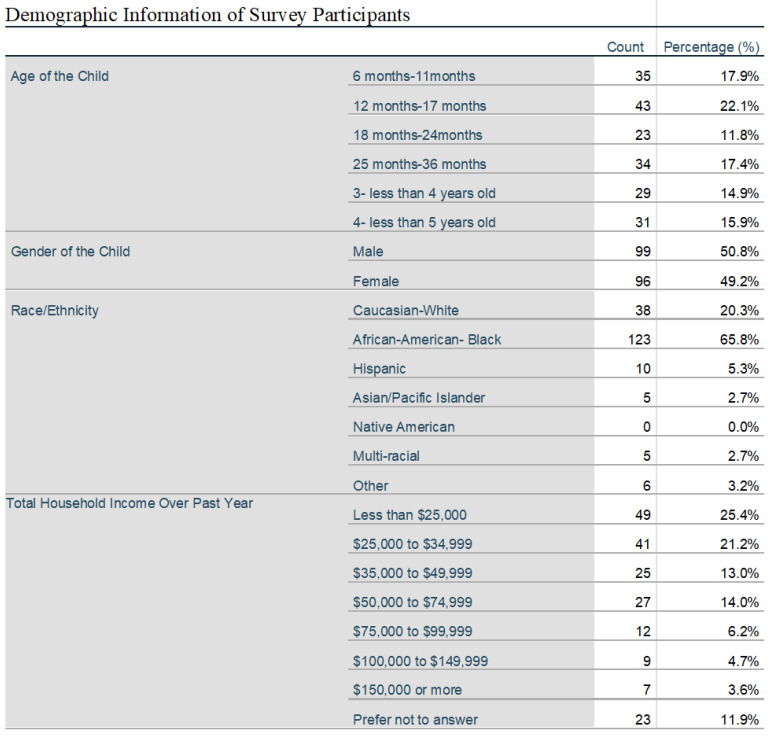
Describes demographic information for individuals who participated in the study from the CHKD GAP clinic.

**Figure 2 vaccines-10-01828-f002:**
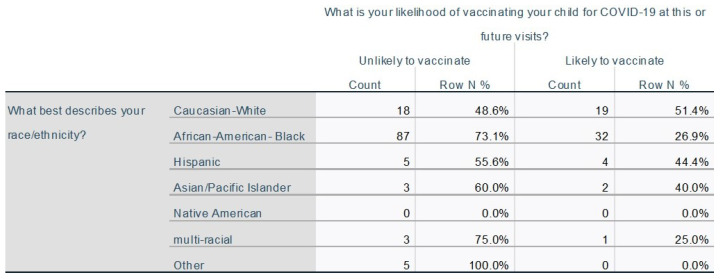
Describes association between an individual’s ethnicity and their likelihood to vaccinate.

**Figure 3 vaccines-10-01828-f003:**
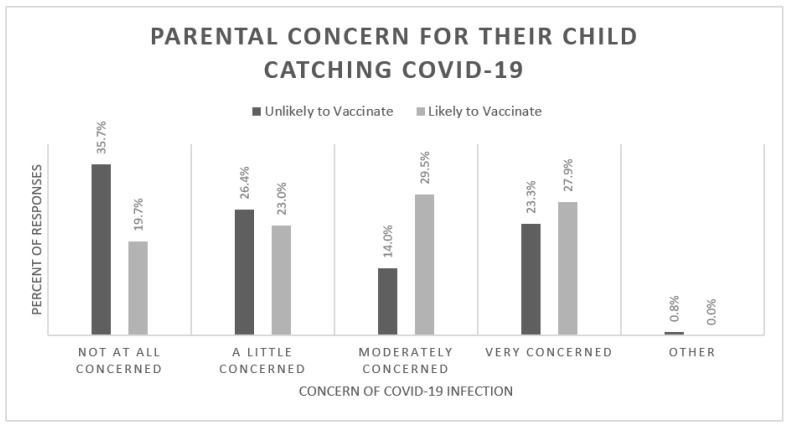
Bar chart displaying the percentage of responses versus parental concern for their child catching COVID-19. There was a statistically significant difference between the two groups (*p* = 0.012).

**Figure 4 vaccines-10-01828-f004:**
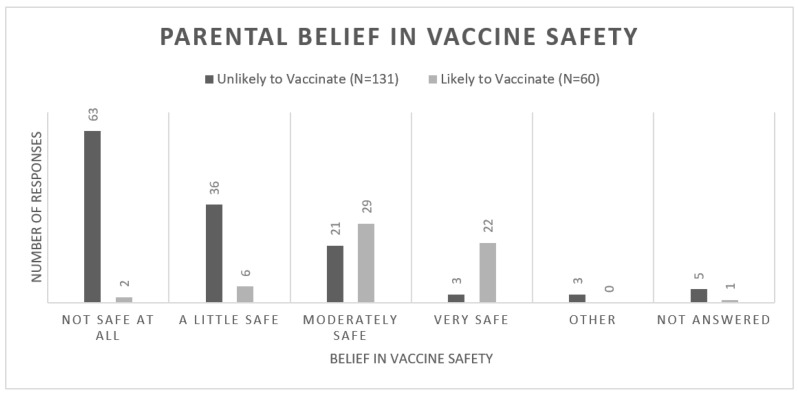
Bar chart displaying the number of responses versus parental belief in COVID-19 vaccine safety. There was a statistically significant difference between the two groups (*p* < 0.001).

## Data Availability

Not applicable.

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
