# Peer review of "Parent’s Hesitation with COVID-19 Vaccinations in Infants and Children Aged 6 Months to 5 Years"

_vaccines, 2022, doi:10.3390/vaccines10111828_

Round 1
Reviewer 1 Report
The data presented are not new at all, but confirm previous information on the topic.
Data ands style presentation is appropriate.
The population sample size is small.
Author Response
Comments and Suggestions for Authors
The data presented are not new at all, but confirm previous information on the topic.
We agree but allows us to look at our specific population that is mostly black
Data ands style presentation is appropriate.
Agree
The population sample size is small.
Agree
Reviewer 2 Report
Authors administered a quantitative, 23-item survey to parents of covid-unvaccinated, 6-59 month old children who presented to pediatric out-patient care in Norfolk VA. The clinic serves mainly patients/parents on Medicaid. Interesting (but not surprising) findings include the inventories of reasons given by participants for or against planned vaccination, including e.g. vaccination would allow resuming social activities or travel (as reasons given for positive vaccination intent), or safety concerns and government mistrust (as reasons given for negative vaccination intent). presentation of results is OK. the discussion of findings and limitations is excellent. methods section is too short.
Specific comments
- abstract: as reader I am disappointed that the authors do not report any results in the abstract. in my view, abstract serves to summarize the major results of the paper. it is not helpful for an abstract to “only” invite the reader to read the paper to get to know the results.
- introduction, line 37: while a UAI may very well be statistically linked to cardiac arrest, in order to avoid alarmism, I recommend authors state that cardiac arrest is an exceedingly rare occurrence in pediatric covid and cite supporting literature.
- introduction, line 37: same for “long term health problems”. authors might consider stating how frequent exactly these are, e.g. long covid and cite supporting literature.
- authors should include the survey in the submission, e.g. in supplementary material
- methods: please state if the survey was available only in English language, or whether other language versions were available.
- methods: along these lines, did researchers ask participants what their preferred language was and how many were non-english speakers ?
- results, lines 87/88: explain what procedure you used to categorize parents into likely vs. unlikely to vaccinate.
- figure 2: consider showing percentages rather than numbers of parents as bars.
- figures 1 and 2: numbers don’t add up to n=131 and n=60, respectively. please explain. perhaps include a “no answer” category ?
- lines 133-138: in its current state, this section on information sources is not so informative. did parents have the option of providing “free text” answers ? or were all answer options pre-supplied by the researchers ?
- line 146: rather than stating that “p>0.05”, please supply the actual p-value, as you did for the statements shown on previous lines.
- discussion, line 153: consider rewording, I recommend “belief” as a more neutral term (this is what parents believe) rather than “misbelief” (which has a negative connotation). as authors appropriately and interestingly state on the following lines 157/158, parents may believe that their kid catching covid is less risky than vaccinating (with a vaccine perceived as unsafe). authors should consider discussing (and providing references) the very appropriate question whether severe covid (unvaccinated) child needs to be hospitalized) is more or less likely than a (vaccinated) child experiencing myocarditis after covid vaccination. such discussions are very frequent with parents and research studies provided varying estimates of the risks of myocarditis vs severe covid.
- some wordings are awkward or hard to understand.
o e.g. line 10 “dosage series” (rec: vaccine series or immunization series).
o line 38: youngest kids are most “social” (rec: are most likely, among all age groups, to have closest and prolonged types of contact with respiratory secretions of other kids)
o line 87: please spell out GED
o line 186: rephrase, i.e. African americans had lower rates OF confidence IN the covid-19 VACCINE (add the word “vaccine”)….
Author Response
Comments and Suggestions for Authors
Authors administered a quantitative, 23-item survey to parents of covid-unvaccinated, 6-59 month old children who presented to pediatric out-patient care in Norfolk VA. The clinic serves mainly patients/parents on Medicaid. Interesting (but not surprising) findings include the inventories of reasons given by participants for or against planned vaccination, including e.g. vaccination would allow resuming social activities or travel (as reasons given for positive vaccination intent), or safety concerns and government mistrust (as reasons given for negative vaccination intent). presentation of results is OK. the discussion of findings and limitations is excellent. methods section is too short.
Specific comments
- abstract: as reader I am disappointed that the authors do not report any results in the abstract. in my view, abstract serves to summarize the major results of the paper. it is not helpful for an abstract to “only” invite the reader to read the paper to get to know the results.
We are limited by 50 words but have tried to convey more information (lines 1-4)
- introduction, line 37: while a UAI may very well be statistically linked to cardiac arrest, in order to avoid alarmism, I recommend authors state that cardiac arrest is an exceedingly rare occurrence in pediatric covid and cite supporting literature.
This has been done(line 36-42)
- introduction, line 37: same for “long term health problems”. authors might consider stating how frequent exactly these are, e.g. long covid and cite supporting literature.
We added information concerning MIS-C (lines 36-42)
- authors should include the survey in the submission, e.g. in supplementary material
This has now been included (lines 249)
- methods: please state if the survey was available only in English language, or whether other language versions were available.
found in (lines 72-76 )
- methods: along these lines, did researchers ask participants what their preferred language was and how many were non-english speakers ?
This is now in lines (72-76)
- results, lines 87/88: explain what procedure you used to categorize parents into likely vs. unlikely to vaccinate.
lines 100-105
- figure 2: consider showing percentages rather than numbers of parents as bars.
This has been changed
- figures 1 and 2: numbers don’t add up to n=131 and n=60, respectively. please explain. perhaps include a “no answer” category ?
Figure has been edited
- lines 133-138: in its current state, this section on information sources is not so informative. did parents have the option of providing “free text” answers ? or were all answer options pre-supplied by the researchers ?
lines 63-66
- line 146: rather than stating that “p>0.05”, please supply the actual p-value, as you did for the statements shown on previous lines.
Value not supplied by statistician
- discussion, line 153: consider rewording, I recommend “belief” as a more neutral term (this is what parents believe) rather than “misbelief” (which has a negative connotation).
line 170
- as authors appropriately and interestingly state on the following lines 157/158, parents may believe that their kid catching covid is less risky than vaccinating (with a vaccine perceived as unsafe). authors should consider discussing (and providing references) the very appropriate question whether severe covid (unvaccinated) child needs to be hospitalized) is more or less likely than a (vaccinated) child experiencing myocarditis after covid vaccination. such discussions are very frequent with parents and research studies provided varying estimates of the risks of myocarditis vs severe covid.
lines 176-185
- some wordings are awkward or hard to understand.
wording has been edited to improve flow
o e.g. line 10 “dosage series” (rec: vaccine series or immunization series).
line 10
- line 38: youngest kids are most “social” (rec: are most likely, among all age groups, to have closest and prolonged types of contact with respiratory secretions of other kids)
- line 42-43
- line 87: please spell out GED
- line 95
- line 186: rephrase, i.e. African americans had lower rates OF confidence IN the covid-19 VACCINE (add the word “vaccine”)….
line 212
Reviewer 3 Report
I found the article interesting and well structured. The introduction provides a useful starting point to motivate the study. The methods are well specified. Instead, I found the exposure of the results unnecessarily long-winded. I suggest a summary of the main results and the addition of a summary table. Well-structured and realistic conclusions. I particularly liked the sentence: " Parents who 161 are ’unlikely to vaccinate’ may not be susceptible to the same type of information that 162 pro-vaccination group are".
I would like to encourage researchers to investigate parents' need of information in order to better target information campaigns.
Author Response
I found the article interesting and well structured. The introduction provides a useful starting point to motivate the study. The methods are well specified. Instead, I found the exposure of the results unnecessarily long-winded. I suggest a summary of the main results and the addition of a summary table. Well-structured and realistic conclusions. I particularly liked the sentence: " Parents who 161 are ’unlikely to vaccinate’ may not be susceptible to the same type of information that 162 pro-vaccination group are".
Results have been edited and streamlined
I would like to encourage researchers to investigate parents' need of information in order to better target information campaigns.
Agreed
Round 2
Reviewer 2 Report
much improved version, my comments have been appropriately addressed.
a few remaining minor issues:
- methods: survey is provided NOT in supplementary materials, as stated on line 66, but as figure 5.
- figure 5. this looks like the entire administered survey, correct ? in this case, the figure legend seems awkward, i.e. here the entire survey is displayed, why should anyone want to request it from the authors via email ?
Author Response
- methods: survey is provided NOT in supplementary materials, as stated on line 66, but as figure 5.
- The wording has been changed to state "A copy of the survey is displayed in Figure 5"
- figure 5. this looks like the entire administered survey, correct ? in this case, the figure legend seems awkward, i.e. here the entire survey is displayed, why should anyone want to request it from the authors via email ?
- Changes to figure 5 caption and the supplementary materials text to were made. I included the option for a pdf upon request in the supplementary materials incase a reader wanted a copy without having to create their own from figure 5.